# Design Optimization and Tradeoff Analysis of an Actuated Continuum Probe for Pulmonary Nodule Localization and Resection

**DOI:** 10.3390/bioengineering11050417

**Published:** 2024-04-24

**Authors:** Madison D. McCullough, Marie Muller, Thomas M. Egan, Gregory D. Buckner

**Affiliations:** 1Department of Mechanical and Aerospace Engineering, North Carolina State University, Raleigh, NC 27695, USA; mdmccul4@ncsu.edu (M.D.M.); mmuller2@ncsu.edu (M.M.); 2Division of Cardiothoracic Surgery, Department of Surgery, University of North Carolina at Chapel Hill School of Medicine, Chapel Hill, NC 27599, USA; thomas_egan@med.unc.edu

**Keywords:** pulmonary nodule, surgical resection, continuum probe, design optimization, genetic algorithm

## Abstract

Pulmonary nodules are abnormal tissue masses in the lungs, typically less than 3.0 cm in diameter, commonly detected during imaging of the chest and lungs. While most pulmonary nodules are not cancerous, surgical resection may be required if growth is detected between scans. This resection is typically performed without the benefit of intraoperative imaging, making it difficult for surgeons to confidently provide appropriate margins. To enhance the efficacy of wedge resection, researchers have developed a modified ultrasound imaging approach that utilizes both multiple scattering (MS) and single scattering (SS) to enhance the accuracy of margin delineation. Clinical deployment of this novel ultrasound technology requires a highly maneuverable ultrasound probe, ideally one that could be deployed and actuated with minimal invasiveness. This study details the design optimization and tradeoff analysis of an actuated continuum probe for pulmonary nodule localization and resection. This device, deployed through intercostal ports, would enable the intraoperative imaging and precise mapping of nodules for improved margin delineation and patient outcomes. To achieve this objective, multiple objective genetic algorithms (MOGAs) and a design of experiments (DOE) study are used to explore the design space and quantify key dimensional relationships and their effects on probe actuation.

## 1. Introduction

Lung cancer is the leading cause of cancer-related death globally, with over 2.2 million new cases and over 1.7 million deaths in 2020, though these rates have both decreased over the previous decade [1]. Clinical imaging of the chest and lungs (using X-ray, computed tomography (CT), positron emission tomography (PET), etc.) is an important tool in the diagnosis and treatment of lung cancer. Such imaging may reveal the presence of pulmonary nodules (PNs): abnormal lung tissue masses typically less than 3.0 cm in diameter [2,3]. PNs are detected in approximately 50% of adults who receive chest X-rays or CT scans, but less than 5% of PNs are cancerous [4,5]. Periodic monitoring of PNs is standard clinical practice: those that exceed size thresholds or grow between subsequent scans are candidates for biopsy or wedge resection [6,7].

### 1.1. Wedge Resection of PNs

Traditionally, wedge resection is accomplished via open thoracotomy, which requires a lateral incision and rib retraction to provide surgical access to the lung [8]. More recently, minimally invasive methods such as video-assisted thoracic surgery (VATS) and robotic-assisted thoracic surgery (RATS) have become common, requiring only small intercostal incisions for the insertion of thoracoscopic instruments. These procedures tend to have less tissue trauma, reduced pain and infection, reduced hospital stays, and have better long-term outcomes [9].

For VATS, the surgical instruments usually include a thoracoscope, which is used to visualize the interior of the thoracic cavity, a linear surgical stapler for tissue resection (Figure 1), and laparoscopic grasping forceps [10,11]. Linear stapler jaws can be opened or closed before the linear cutter is launched to enable resection between rows of stapled tissue for minimal blood loss [12]. RATS uses similar instruments mounted to the arms of a tele-operated robotic surgery system. Though there exists no global surgical standard for RATS operations, surgeons commonly use a four-port technique with the seventh and eighth intercostal spaces most commonly used for instrument port placements as shown in Figure 2 [13].

Despite the benefits of preoperative CT and PET scans, these imaging modalities are generally not used intraoperatively. Furthermore, ultrasound is not considered a viable method for lung imaging as the lung’s air-filled alveoli introduce air–liquid interfaces that cause the significant multiple scattering (MS) of ultrasound waves. MS occurs when a wave experiences multiple reflections before returning to the transducer, as opposed to single reflection events, defined as single scattering (SS) [14]. MS introduces nonlinearity to the relationship between propagation time and the distance from the transducer due to wave distortion and absorption.

As a result, preoperative scans are frequently used in conjunction with intraoperative localization techniques, including scan-guided percutaneous insertion or the placement of hookwires, microcoils, medical dyes, or medical adhesives into the area surrounding the nodule [9,15,16,17,18,19]. Although these techniques help pinpoint the locations of PNs intraoperatively, they can be adversely affected by marker migration and have been associated with increased risk of pneumothorax, increased healthcare costs, and increased surgical invasiveness [9].

The lack of intraoperative imaging, combined with the near impossibility of palpation via surgical instruments, makes it difficult for surgeons to confidently resect pulmonary nodules with appropriate margins. This can result in surgeons upgrading minimally invasive procedures to open thoracotomies; this occurs in 8.8% of VATS resections [20], 19.2% of VATS lobectomies, and 11.5% of RATS lobectomies [21]. Margin delineation can be considerably more difficult for nodules with less defined boundaries. As a result, some surgeons opt for larger margins to better ensure the nodule is entirely contained within the resected tissue. If the entirety of the nodule is not successfully resected, the chance of cancer recurrence increases [9].

For these reasons, research by Roshankhah, et al. [14] introduced a modified ultrasound imaging approach to pulmonary nodule localization that utilizes both multiple scattering (MS) and single scattering (SS) to enhance the effectiveness of margin delineation via real-time imaging. Because alveoli do not exist in pulmonary nodules, MS is reduced significantly in comparison to adjacent lung tissue. By segregating backscattered waves into SS and MS, maps of the SS/MS ratio provide contrast between pulmonary nodules and normal tissue and can be used for the intraoperative identification and localization of PNs (Figure 3). This technology may also be postoperatively used to verify that the entire nodule is contained within extracted resected tissue.

Such a device, deployed through intercostal ports, would enable the intraoperative imaging and precise mapping of nodules for improved margins and improved patient outcomes. 

### 1.2. Probe Actuation

The intraoperative use of ultrasound SS/MS mapping requires a highly maneuverable ultrasound probe, ideally one that could be deployed and actuated with minimal invasiveness (i.e., through intercostal ports). This probe must possess the capability to precisely traverse the lung exterior, maneuvering and rotating such that complete tangential contact between the ultrasound transducer array and the lung tissue is maintained.

Actuation options were determined from an extensive literature review of similar actuated surgical instruments. Such devices typically utilize a continuous or articulated central structure for axial stiffness and drive cables for lateral bending actuation [22]. Examples include continuum structures with articulated ball and socket joints [23,24,25] or flexible central beams [26], origami-based graspers [27], and concentric tube mechanisms [28,29]. While concentric tube mechanisms provide excellent axial stiffness and navigational precision, they are generally not capable of large bending deflections and lack a sufficiently large central lumen to route the transducer’s large number of electrical conductors. While flexible beam options also provide excellent axial stiffness with much larger bending deflections, they too lack a sufficient inner lumen diameter for probe electrical leads. Origami-based mechanisms generally lack axial stiffness and are functionally complicated with overly large outer diameters. Continuum ball and socket structures were selected for this optimization study because of their inherent axial stiffness and lateral bending capabilities and their relatively large inner lumen diameters. These characteristics are important for precise maneuvering within the thoracic cavity and maintaining tangential probe contact with lung tissues during imaging.

A continuum ball and socket structure is composed of vertebral elements that articulate via tensioned cables connected to the ultrasound probe. These tendons also constrain axial translation and rotation. Each vertebra features a central lumen for the probe’s electrical leads as well as four equidistant radial lumens for the actuating tendons.

### 1.3. Research Objective

The primary objective of this study was to analyze tradeoffs in the design of a continuum ball and socket actuator and optimize it for ultrasound probe positioning during pulmonary nodule resection. To achieve this objective, multiple objective genetic algorithms (MOGAs) and design of experiments (DOE) were used to explore the design space and quantify relationships between key vertebra dimensions and their effects on probe actuation.

## 2. Materials and Methods

The ultrasound transducer for this study was harvested from a commercial laparoscopic ultrasound instrument, the Aloka UST-5536-7.5 Intraoperative Electronic Linear Probe pictured in Figure 4 [30]. This manually actuated ultrasound probe is commonly used for the intraoperative imaging of solid organs (e.g., the liver, pancreas, kidneys, etc.) and to facilitate minimally invasive procedures (e.g., tissue biopsy, fiducial marker placement, ablation, and gynecological surgery) [31]. This probe’s linear transducer has a 10.0 mm diameter and a length of approximately 60 mm, imposing critical design constraints on the tradeoff study [32]. This transducer was mounted to the articulated probe body, its electrical conductors routed through the central lumen, and it was used in conjunction with the novel SS/MS scattering ratio technique to enable the real-time mapping of nodules for improved resection margins.

To establish kinematic and design requirements for the actuated probe, CT scan data of an adult human thoracic cavity [33] were used to create a point cloud model in MATLAB (Figure 5). This model was scaled based on published anthropometric measurements to provide the 40% deflation common in lung resections [34]. Simulated instrument port locations P1-7 (three on the patient’s left side, four on the right) were based on a survey of high-volume thoracic surgeons [13], with the fourth right side port representing an alternative posterior port for the right lower lung. Simulated pulmonary nodule locations PN1-10 (two for each of the five lung lobes: the right upper, right middle, right lower, left upper, and left lower) were similarly chosen based on the published literature to represent a wide range of surgical conditions.

### 2.1. Probe Configuration Analysis

Figure 6 shows the deployment of a straight probe body through the 2nd instrument port on the patient’s right side. Because each port (and associated trocar) has a larger inner diameter than the probe’s outer diameter, rotational degrees of freedom enable a cone-shaped reachable space for the probe shaft; it can be rotated and pivoted between −*θ_c_* and +*θ_c_* from the port’s normal unit vector ${\hat{u}}_{P_{2}}$, which was determined from the anatomical point cloud using MATLAB’s pcnormals command.

To identify the optimal probe configurations for imaging each target pulmonary nodule, namely the configurations that provide the shortest paths from the instrument ports to the target pulmonary nodule, geometric optimization was conducted. For each nodule, proximal line segment lengths ($L_{p}$) and distal segment lengths ($L_{d}$) were computed for 50 different offset angles *θ_c_* for each port on the corresponding side (Figure 7). The proximal segment originated from the *ith* port center ($x_{pi},\;y_{pi},\;z_{pi}$) (within *±*45° of the centered port normal vector) and extended into the pleural space, while the distal segment lay on the plane tangent to the lung surface at the target pulmonary nodule ($x_{nj},\;y_{nj},\;z_{nj}$). Using MATLAB’s *pcnormals* command, unit vectors normal to the lung surface through each of the *j* nodules were constructed. These normal unit vectors were used to define plane tangents to the lung surface for each nodule. These planes were then used to define distal line segments originating from the surface normal to the intersection points with the proximal line segments. The optimal probe configurations for each port were those that minimized the total probe length ($L_{p}$ + $L_{d}$), as shown in Figure 8.

### 2.2. Design Variables and Constraints

The proposed actuated continuum probe for pulmonary nodule localization and resection features a series of articulated vertebrae, parameterized by the dimensions shown in Figure 9 and summarized in Table 1. For design optimization, dimensions $D_{3}$, $D_{5}$, $t_{i}$, and $t_{o}$ were fixed as each electrical lead of the Aloka UST-5536-7.5 transducer has a 0.4 mm diameter; each actuating cable was also assumed to have a 0.4 mm diameter. Clearances of 0.5 mm for the central lumen and 0.2 mm for the radial lumen enabled smooth relative motion between the cables and lumens. The outer wall thickness *t_o_* must be sufficiently thick to prevent wall failure.

Upper and lower parameter bounds were specified based on surgical and anatomical dimensions and probe dimensional constraints to facilitate the complete exploration of the design space. For example, the upper bound on spherical radius $R_{2}$ was based on typical surgical port dimensions and its kinematic relationship with central lumen diameter $D_{3}$, while socket depth μ was minimally bounded to ensure vertebral mating and was limited by $R_{2}$. Some design parameters depended directly on others: radial lumen dimension $D_{4}$ depended on outer flange diameter $D_{1}$, and lumen diameter $D_{5}$ and outer wall thickness, $t_{o}$:(1)$${D_{4}=D}_{1}-2\;\left(t_{o}\right)-D_{5}.$$

Linear and nonlinear design constraints were implemented to prevent infeasible and unfavorable geometries, as detailed in [35]:(2)$$H_{2}-H_{1}\leq 0,$$
(3)$$H_{1}-2R_{2}\leq 0,$$
(4)$$\mu +H_{1}\leq -0.1,$$
(5)$$\mu -R_{2}\leq c$$
(6)$$∆-\frac{R_{2}}{2}\leq 0,$$

An additional constraint was imposed to ensure adequate space (5.0 mm) for the probe’s electrical leads within the central lumen during extreme actuation.

### 2.3. MOGA-Based Design Optimization

A multiple objective genetic algorithm (MOGA) was used for the design tradeoff analysis based on its suitability to nonlinear, multimodal problems with many design parameters. MATLAB’s *gamultiobj* command, which is based on NSGA-II [36], was used for MOGA-based design optimization. For this study, the population size was set to 50 with a maximum generation number of 40. The number of elite children was set to 3, the crossover rate was set to 0.8, and the remaining population was subject to mutation. At each design iteration, *gamultiobj* generated a seven-variable design string, which was written to a Microsoft Excel design table.

The bending characteristics of each candidate design (set of probe dimensions) was analyzed using a 3D solid modeling application, SolidWorks (Dassault Systèmes, Vélizy-Villacoublay, France). SolidWorks enabled 3D visualization and kinematic analysis from the probe’s minimum (unbent) initial configuration to its maximum (fully bent) configuration when vertebra-to-vertebra contact was first made. SolidWorks macros were used to assemble vertebrae to a minimum length of 55 mm for consistent evaluations of the performance objectives.

Three performance objectives were formulated to maximize the surgical utility of the actuated ultrasound probe during design optimization: (1) minimizing the radius of curvature at maximum deflection to improve reachability, (2) minimizing the mass moment of inertia for enhanced bandwidth and maneuverability, and (3) minimizing the actuator’s outer diameter to reduce invasiveness. For each candidate design, the probe’s radius of curvature, mass moment of inertia, and outer diameter were computed using SolidWorks. An arc connecting the spherical centers of each vertebra at maximum probe deflection was used to quantify the radius of curvature ρ, the first performance objective value. The mass moment of inertia for each design (about the probe’s *x* axis, $I_{xx}$) was calculated by resolving the vertebrae into a series of equally spaced point masses and employing the parallel axis theorem:(7)$$I_{xx}=m_{1}{(d}_{1}-r_{1})+\sum_{i=2}^{n}{\;\left(m_{i}d_{i}\right)}^{2}$$
where $m_{i}$ is the mass of an individual vertebrae (computed by SolidWorks based on AISI Type 316L stainless steel material), $d_{i}$ is the distance between adjacent spherical centers, and $r_{1}$ is the distance from the actuator base to the spherical center of the first deflecting vertebra. This calculation was performed for an undeflected assembly and was verified using the SolidWorks Mass Properties Moments of Inertia calculator. All three performance objectives were reported to the SolidWorks Equation Manager and written to the Excel results file.

If a generated design violated any of the nonlinear constraints, an arbitrarily large value of 10,000 was assigned to each objective to reduce the chances of its genetic information being passed to the next generation and for later identification during data analysis. After the optimization process was complete, the few remaining designs that violated nonlinear constraints were simply excluded from further analysis.

## 3. Results

### 3.1. Optimal Probe Configuration

The results of kinematic simulations (Table 2) show that the nodules located in the upper lobes experience the shortest probe paths from more anteriorly located instrument ports, and those in the lower lobes experience the shortest probe paths from more posteriorly located instrument ports. Instrument Port 1 typically has longer proximal segments, while Port 3 has shorter combined lengths. The required radii of curvature are generally only a few centimeters.

### 3.2. Design Optimization

Three separate MOGA trials were conducted with 40 design generations each to confirm the repeatability of the validation process. Each trial resulted in 18 individuals on the Pareto frontier; examples are provided in Figure 10. Infeasible designs, namely ones that violate nonlinear constraints, were identified using Microsoft Excel data filters and removed from the results. An example is shown in Figure 11, where an inadequate flange width was created for the radial lumen, violating the associated nonlinear constraint.

The design space for one trial is displayed in Figure 12, and the Pareto frontiers for each trial are presented in Figure 13.

Figure 13 shows that the Pareto frontiers for each trial are relatively similar. The mass moment of inertia vs. the radius of curvature plot (Figure 13 top left) and the outer diameter vs. the radius of curvature plot (Figure 13 top right) both exhibit nonlinear relationships. This indicates that the outer diameter and mass moment of inertia objectives conflict with the radius of curvature objective as further highlighted in the parallel coordinates plot of Figure 14. The same was not true for the mass moment of inertia vs. the outer diameter, however, as this plot (Figure 13 bottom) featured a more positive linear trend. Typically, a lower value of outer diameter signifies a lower value of the moment of inertia.

For Trial 1 and Trial 2, the designs with the smallest mass moment of inertia and the smallest diameter have the same design string as shown in Table 3. The smallest radius of curvature values for Pareto-optimal designs have some of the largest mass moments of inertia and outer diameters as shown in the last three columns of the table for the three objectives. Table 3 also shows that Pareto-optimal designs with the smallest radii of curvature for each trial have higher $R_{2}$, Δ, λ, and μ values in comparison to designs with the smallest mass moment of inertia and outer diameter, potentially displaying the significance of these design variables in objective function values. To further investigate these relationships and the effects of other design variables, the variables of every design in the space are plotted against the three objective functions in Appendix A.

These plots reveal that $R_{2}$ takes on a somewhat even range of values across the design space which was further displayed along the Pareto front, with color transitions in Appendix A. Lower $R_{2}$ values are associated with lower moments of inertia and outer diameters, which supports the results of Table 3. However, higher values of $R_{2}$ do not necessarily result in lower radius of curvature values. Δ also has an even spread, though for only about half the design space, which could be attributed to MOGA constraints. Very low Δ and λ values are associated with the designs optimal with respect to mass moment of inertia and outer diameter, while mid-range values are associated with minimal radius of curvature designs. Mid-range μ values are common close to the frontier as well as the intermediate ranking designs; low-range μ values are less common and are associated with some of the lowest moment of inertia and outer diameter designs, consistent with previous conclusions.

The design space reveals that candidates with low $H_{1}$ values, low $H_{2}$ values, and high ψ values typically have low moments of inertia and small outer diameters. However, the range of $H_{1}$ and $H_{2}$ was mixed for low radii of curvature as opposed to ψ, which tended toward lower values for lower radii of curvature.

Multivariate analysis of variance (MANOVA) was used to quantify the effects of the seven design variables on the three objective functions using the IBM^®^ SPSS^®^ Statistics program (IBM, Armonk, NY, USA). As shown in the results of Table 4, each MANOVA had a multivariate *p*-value less than a set significance level of 0.05, indicating that changing the level of each design variable, without altering others, provided statically significant differences on the multivariate response. Variables ψ, λ, and $H_{1}$ had the greatest ${\eta_{p}}^{2}$ values, signifying that they had the greatest effects on the multivariate response. $R_{2}$ and $H_{2}$ had some of the least effect. Observing the univariate results, it can be seen that *ψ* and $H_{2}$ had no significant effect on the radius of curvature response; Δ, λ, and μ had no significant effect on the moment of inertia response; and Δ and μ had no significant effect on the outer diameter response. λ had the greatest effect on the radius of curvature response’s variation followed by $H_{1}$ and Δ, with changes in their values attributing to 44.5%, 17.1%, and 12.8% of the variation respectively. For the moment of inertia and outer diameter responses, $H_{1}$ and ψ had the greatest effect on both.

### 3.3. Experimental Confirmation

In order to confirm the actuation capabilities of the MOGA-based design optimization, two Pareto-optimal designs from Trial 1 were chosen for prototyping and experimental analysis. Specifically, the Pareto-optimal designs with the smallest radius of curvature (Model 1) and the lowest moment of inertia and outer diameter (Model 2) were 3D printed in clear resin using a Formlabs Form 3+ SLA printer with enough copies made to create a stacked length of 100 mm (Figure 15). This required 12 vertebrae for Model 1 and 17 vertebrae for Model 2, which were then assembled with the actuating wire for accurate comparison.

Each prototype was connected to the benchtop test setup pictured in Figure 16, which featured an Arduino Mega 2560 Rev 3 microcontroller for actuating four DC gearmotors (Pololu 380:1 Micro Metal Gearmotor HPCB 12V) and pulley mechanisms that varied actuating wire lengths in response to joystick inputs. The system was powered by a Keysight E3620A DC Power Supply.

Both models were fastened to the experimental setup and actuated to their maximum deflected state in a single plane. Model 1 had an experimental radius of curvature of approximately 20 mm, which compared favorably to its theoretical radius of curvature of 19.94 mm (Figure 17). Similarly, Model 2 had an experimental radius of curvature of approximately 40 mm, which compared less favorably to its theoretical radius of curvature of 55.87 mm (Figure 18). This discrepancy between the theoretical and experimental radius of curvature is likely attributed to variations in clearance spacing between vertebrae. It is important to also note that printing and assembly errors could contribute to discrepancies in the reported experimental values, but the overall outcome demonstrates the feasibility of using these kinds of vertebrae for pleural cavity navigation. Although Model 1 demonstrated superior bending capabilities, its greater outer diameter could be associated with greater surgical invasiveness compared to Model 2.

## 4. Discussion

Both the MOGA-based optimization and the DOE provided opportunities to quantify, visualize, and interpret key tradeoffs in the design of a continuum ball and socket actuator for ultrasound probe positioning during pulmonary nodule resection. Such a device, deployed through intercostal ports, would enable intraoperative imaging and precise mapping of nodules for improved margins and improved patient outcomes. The MOGA was effective in determining designs optimal in radius of curvature, mass moment of inertia, and outer diameter. The DOE effectively quantified tradeoffs associated with key design variables and could be used in future work to fine-tune performance objective responses. These analyses reveal that designs optimal for mass moment of inertia and outer diameter are generally not optimal for radius of curvature. The experimental analyses confirmed the accuracy of computational optimization and clinical feasibility of a continuum ball and socket actuator for ultrasound probe positioning during pulmonary nodule resection. Improvements in sensing capabilities and controller sophistication would help improve the accuracy and commercial viability of such a device.

The computational tools used in this study, multiple objective genetic algorithms (MOGAs) and design of experiments (DOE) have inherent assumptions and limitations that could affect the accuracy and robustness of the dimensional relationships. The inherent assumptions of continuous and convex objective spaces, the potential for convergence to local minima, and the unknown sensitivities of design outcomes to variations in design parameters could adversely impact the accuracy of results and conclusions presented here. This study was computationally limited by the maximum number of MOGA generations, population size, and number of trials conducted. In future work, it might be necessary to vary crossover rates, mutation rates, population sizes, and stopping criteria; all clearly affect the convergence of Pareto frontiers and the complete exploration of the design space. Also, using improved methods for evaluating the convergence of the multiple objective design optimization, such as hypervolume metrics, might be beneficial. A more sophisticated anatomical simulation could also be used to better quantify the probe actuation requirements by including additional constraints caused by the introduction of other laparoscopic instruments (staplers, cameras, etc.) into the surgical space. This simulation could also be used to observe whether having multiple bending segments would be beneficial for probe navigation.

Future research could explore different continuum joint designs and performance objectives, which could be compared to the baselines presented here. One such performance objective could include an actuation force analysis to determine the effects of varying forces that a laparoscopic probe would be subject to during the steps of thoracic surgery. Also, future work could include a DOE with a greater number of levels to provide greater statistical significance.

## Figures and Tables

**Figure 1 bioengineering-11-00417-f001:**
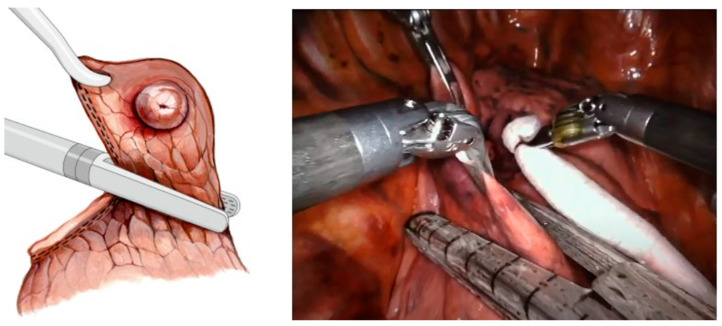
Linear surgical staplers during pulmonary nodule resection. (**Left**) illustrated surgical technique for peripheral nodule (reprinted with permission from Ref. [10], the European Society of Thoracic Surgeons, 2023); (**right**) thoracoscope’s view of Endo GIA^TM^ stapler used in conjunction with Da Vinci EndoWrist^TM^ fenestrated graspers during a RATS wedge resection [11].

**Figure 2 bioengineering-11-00417-f002:**
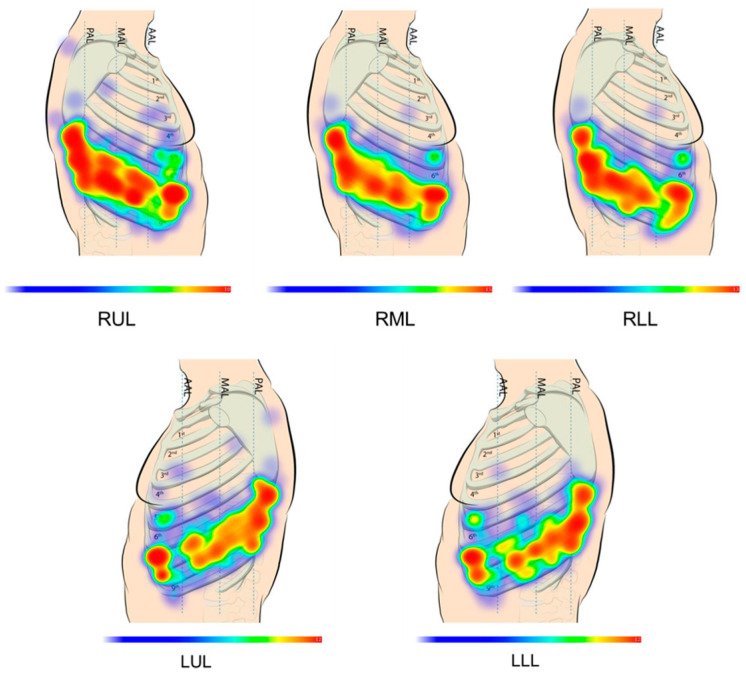
Heat color map displaying instrument port placement frequency based on survey responses of U.S.-based, high-volume robotic thoracic surgeons for four-arm technique RATS for pulmonary nodules located in the right upper lobe (RUL), right middle lobe (RML), right lower lobe (RLL), left upper lobe (LUL), and left lower lobe (LLL). Red indicates higher frequency port placement, while dark blue indicates least frequent locations [13].

**Figure 3 bioengineering-11-00417-f003:**
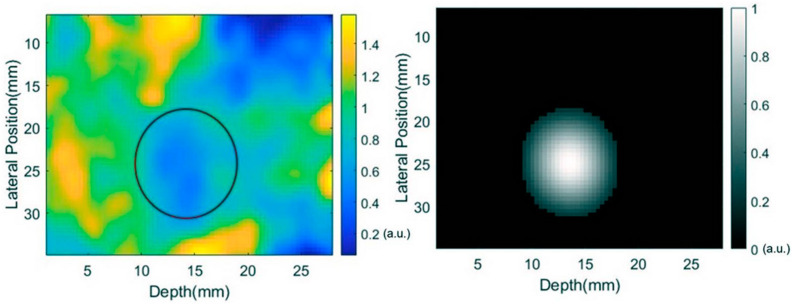
SS/MS ultrasound technique for pulmonary nodule detection. (**Left**) normalized SS intensity standard deviation map of ex vivo swine lung injected with nodule material using 128-element linear transducer array with central frequency of 5.2 MHz. (**Right**) rendered map using depression detection algorithm to quantify the standard deviation of SS intensity, showcasing nodule location in white. Reproduced from [14], with the permission of the Acoustical Society of America.

**Figure 4 bioengineering-11-00417-f004:**
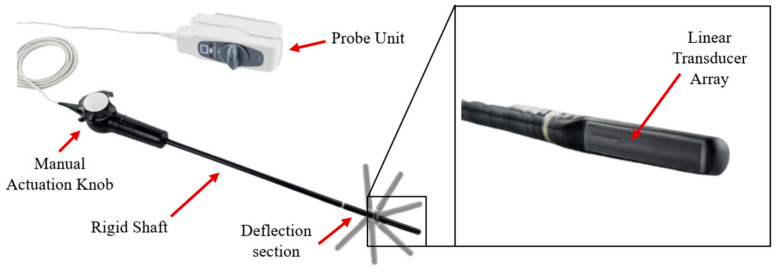
Aloka probe system with key components parts labeled. This probe has a 10 mm diameter, 380 mm long insertable shaft consisting of a 250 mm rigid section connected to the operating head, an ~60 mm deflection section, and an ~60 mm tip featuring the linear transducer [30].

**Figure 5 bioengineering-11-00417-f005:**
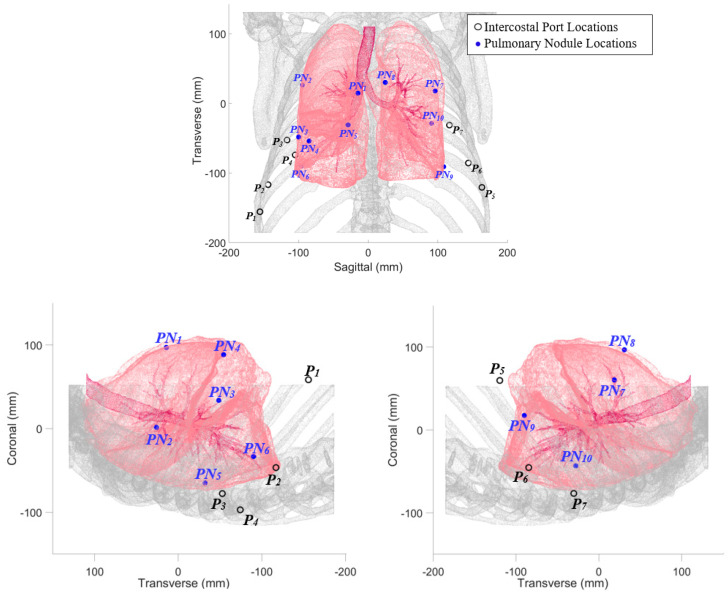
Point cloud model of human adult thoracic cavity, with partially deflated lungs shown in pink, ribs in gray [33]. (**Top**): coronal plane view; (**bottom left**): sagittal plane view of right lung; (**bottom right**): sagittal plane view of left lung. Intercostal instrument ports P1-7 are based on a survey of high-volume thoracic surgeons [13] and pulmonary nodule locations. PN1-2 are located on the right upper lobe, PN3-4 are located on the right middle lobe, PN5-6 are located on the right lower lobe, PN7-8 are located on the left upper lobe, and PN9-10 are located on the left lower lobe.

**Figure 6 bioengineering-11-00417-f006:**
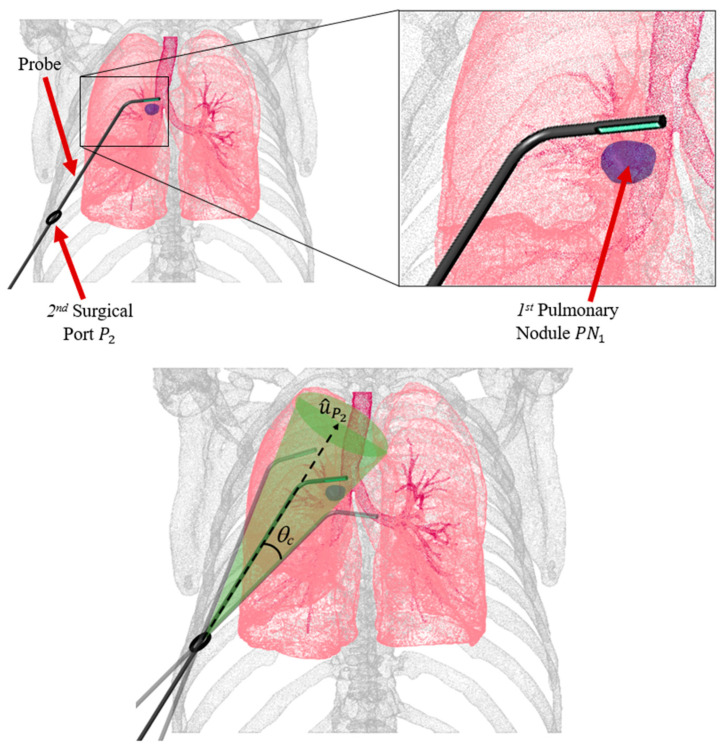
(**Top**) Thoracic cavity with 2nd surgical instrument port *P*_2_ and associated normal unit vector ${\hat{u}}_{P_{2}}$, actuated probe, and 1st pulmonary nodule *PN*_1_ located anteriorly in the right upper lobe. (**Bottom**) probe’s rotational degree of freedom represented by a cone bounded by *θ_c_*.

**Figure 7 bioengineering-11-00417-f007:**
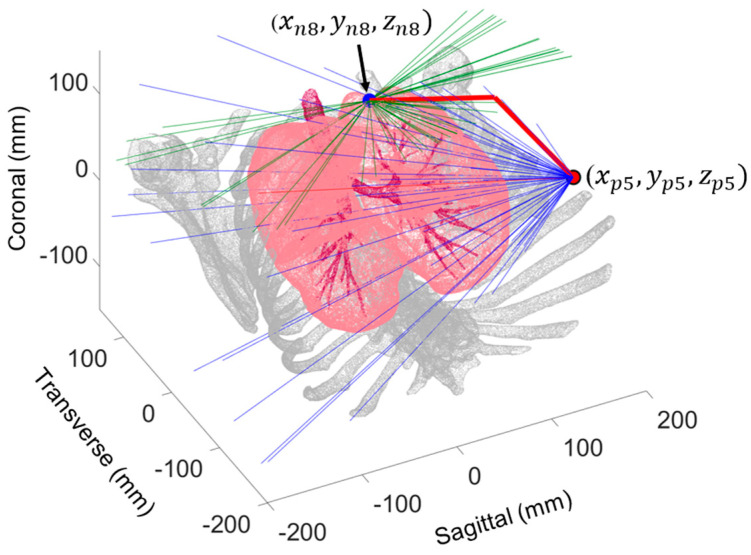
Simulated probe configuration for 5th surgical instrument port ($x_{p5},\;y_{p5},\;z_{p5}$) imaging 8th nodule ($x_{n8},\;y_{n8},\;z_{n8}$) with green surface tangent lines and blue port vectors. The bold red lines represent the shortest path.

**Figure 8 bioengineering-11-00417-f008:**
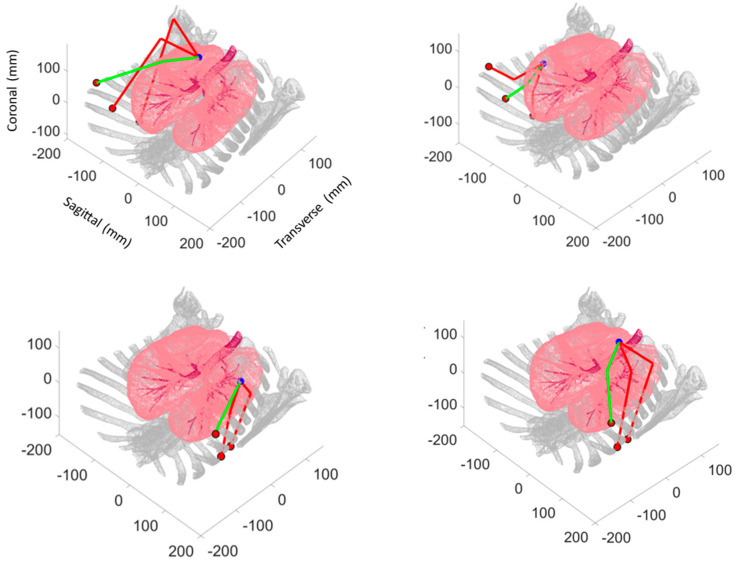
Optimal probe configurations (intersecting proximal and distal line segments) for each port location. Thick green lines designate overall minimum, thick red lines represent secondary minimums for: (**top left**) ${PN}_{1}$, located in right upper lobe; (**top right**) ${PN}_{3}$, located in right middle lobe; (**bottom left**) ${PN}_{7}$, located in left upper lobe; and (**bottom right**) ${PN}_{8}$, located in left lower lobe.

**Figure 9 bioengineering-11-00417-f009:**
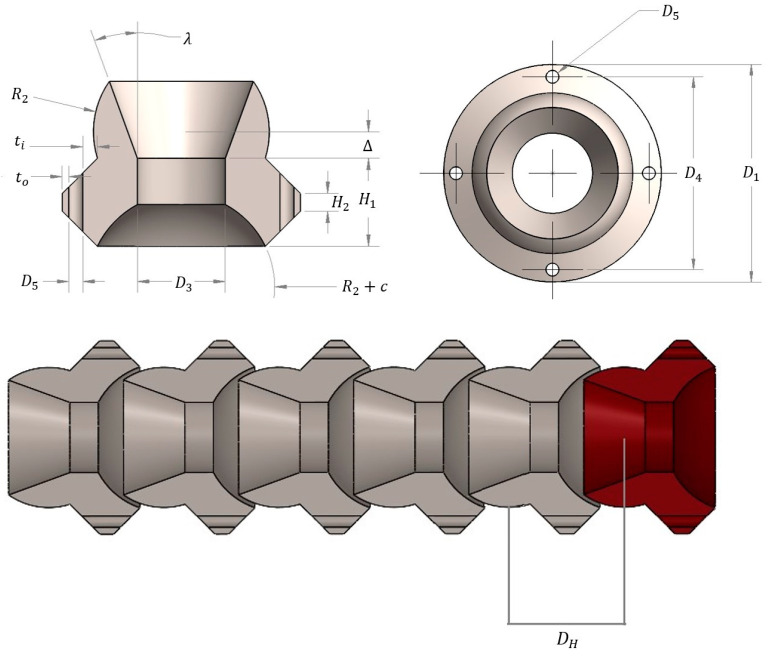
Mechanical structure of actuated continuum probe for pulmonary nodule localization and resection. (**Top left**) top view of parameterized with dimension labels for $R_{4}$, $D_{1}$, and $D_{5}$. (**Top right**) side view of parameterized vertebra with dimension labels for $R_{2}$, $H_{1}$, $H_{2}$, $D_{3}$, λ, μ*,*
$\psi ,\;t_{i}$, $t_{o}$, and Δ. (**Bottom**) serially linked vertebrae with red vertebral base unit with the distance between spherical centers $D_{H}$.

**Figure 10 bioengineering-11-00417-f010:**
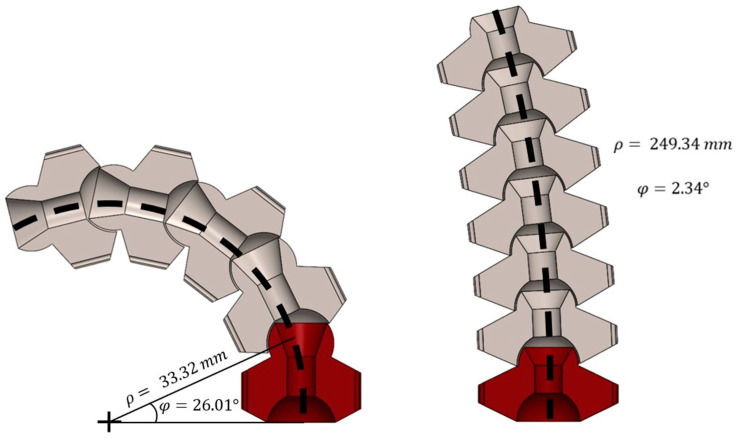
MOGA-generated Pareto-optimal designs rendered at maximum deflection in SolidWorks: (**left**) with the lowest maximum deflection angle of φ=2.34° and ρ=249.34 mm for Trial 1; (**right**) with the greatest maximum deflection angle of φ=26.01° and ρ=33.32 mm for Trial 1.

**Figure 11 bioengineering-11-00417-f011:**
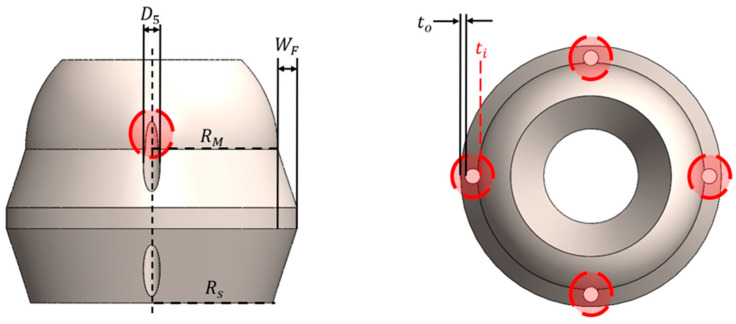
MOGA-generated design that violated nonlinear flange width constraint with design variables: $R_{2}=6.00$ mm, ∆=0.01 mm, $H_{1}=7.30$ mm, ψ=71.95°, $H_{2}=1.00$ mm, λ=22.44°, and μ=3.73 mm. Inadequate flange space is provided for the radial lumen where $R_{S}+D_{5}+t_{o}+t_{i}-W_{F}-R_{M}>0$, so the lumen run partially through the spherical part of the vertebra as $t_{i}$ is nonexistent.

**Figure 12 bioengineering-11-00417-f012:**
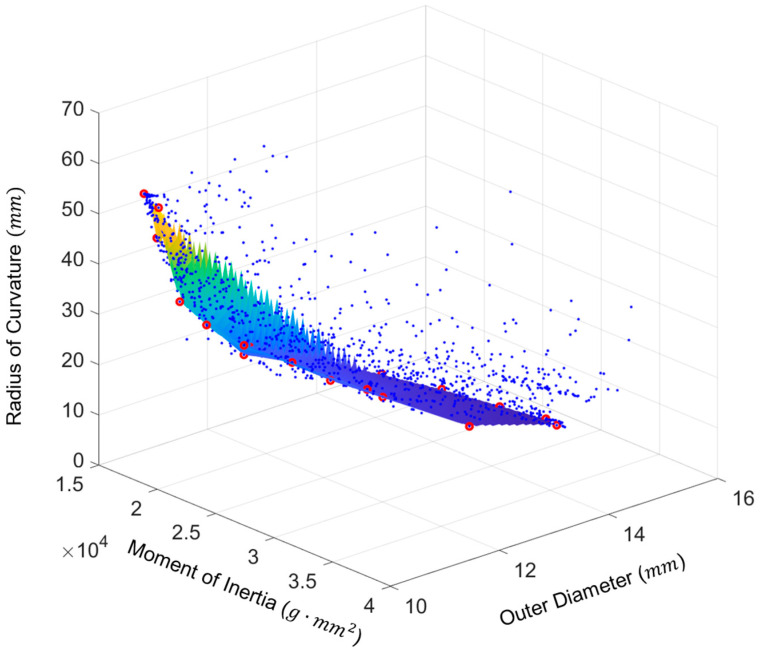
MOGA Trial 1 design space showing feasible-dominated designs in blue and Pareto-optimal designs in red. A scattered interpolant plot was fit to the Pareto front to serve as a possible reference for other high-ranking designs not tested in this algorithm.

**Figure 13 bioengineering-11-00417-f013:**
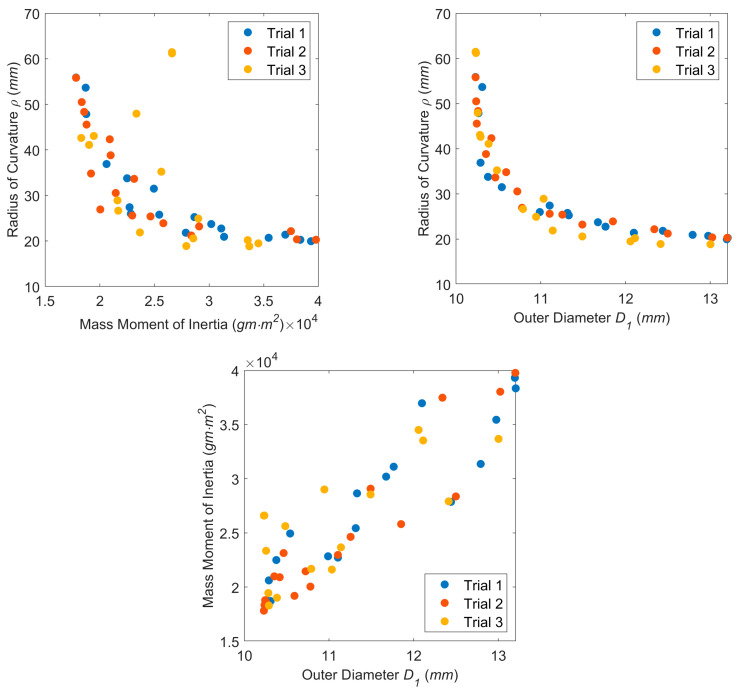
Pareto frontiers for three MOGA trials (40 generations each): (**top left**) mass moment of inertia vs. radius of curvature; (**top right**) outer diameter vs. radius of curvature; (**bottom**) outer diameter vs. mass moment of inertia.

**Figure 14 bioengineering-11-00417-f014:**
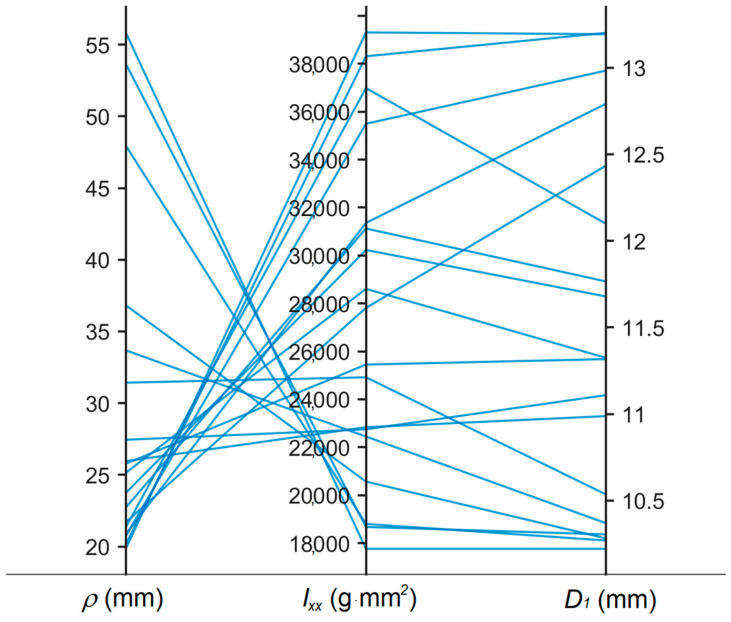
Parallel coordinates plot of the three objectives for Trial 1 displaying the conflicting nature between radius of curvature and other two objectives.

**Figure 15 bioengineering-11-00417-f015:**
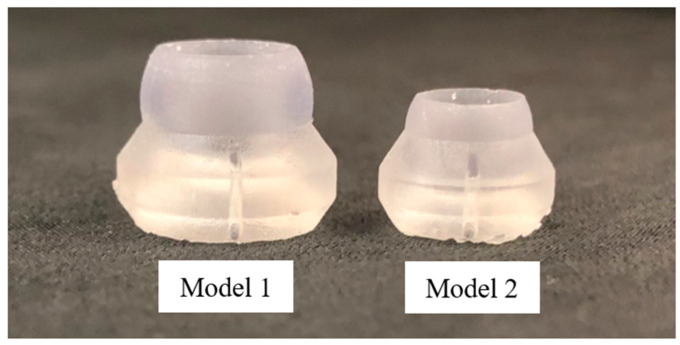
3D printed prototypes of Pareto-optimal vertebrae: (**left**) Model 1 with the smallest radius of curvature and dimensions: $R_{2}=5.14$ mm, ∆=1.28 mm, $H_{1}=6.18$ mm, ψ=54.07°, $H_{2}=2.29$ mm, λ=20.12°, and μ=4.75 mm; (**right**) Model 2 with the lowest mass moment of inertia and dimensions: $R_{2}=3.66$ mm, ∆=0.01 mm, $H_{1}=5.93$ mm, ψ=58.31°, $H_{2}=1.72$ mm, λ=15.00°, and μ=3.97 mm.

**Figure 16 bioengineering-11-00417-f016:**
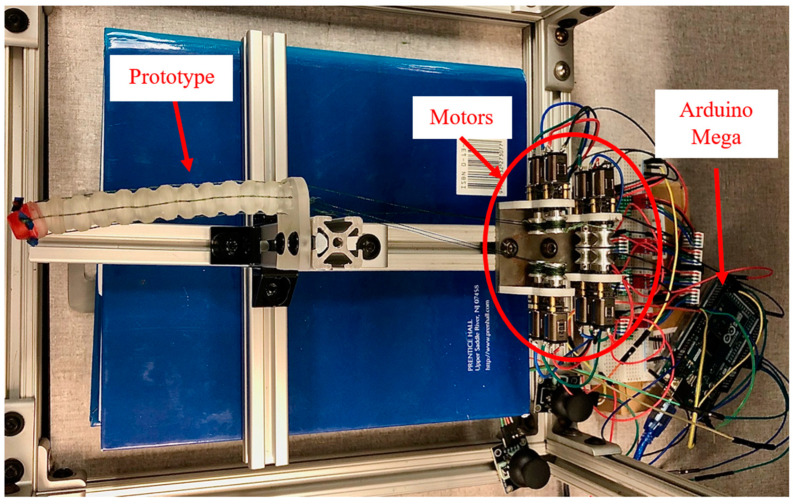
Experimental setup for prototype testing, where four independent motors are used to move the prototype up, down, left, and right.

**Figure 17 bioengineering-11-00417-f017:**
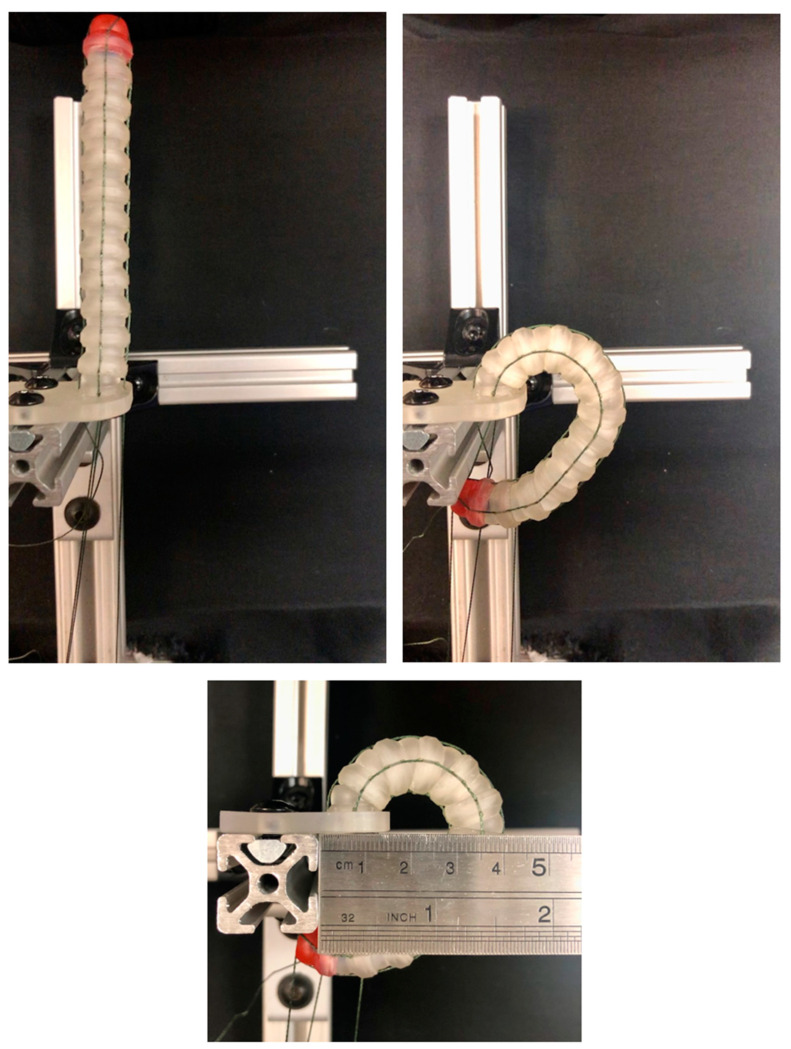
Photographs of experimental validations of Model 1 prototype: (**top left**) in undeflected state; (**top right**) in maximum deflection state; (**bottom**) with ruler showing a radius of curvature of approximately 20 mm.

**Figure 18 bioengineering-11-00417-f018:**
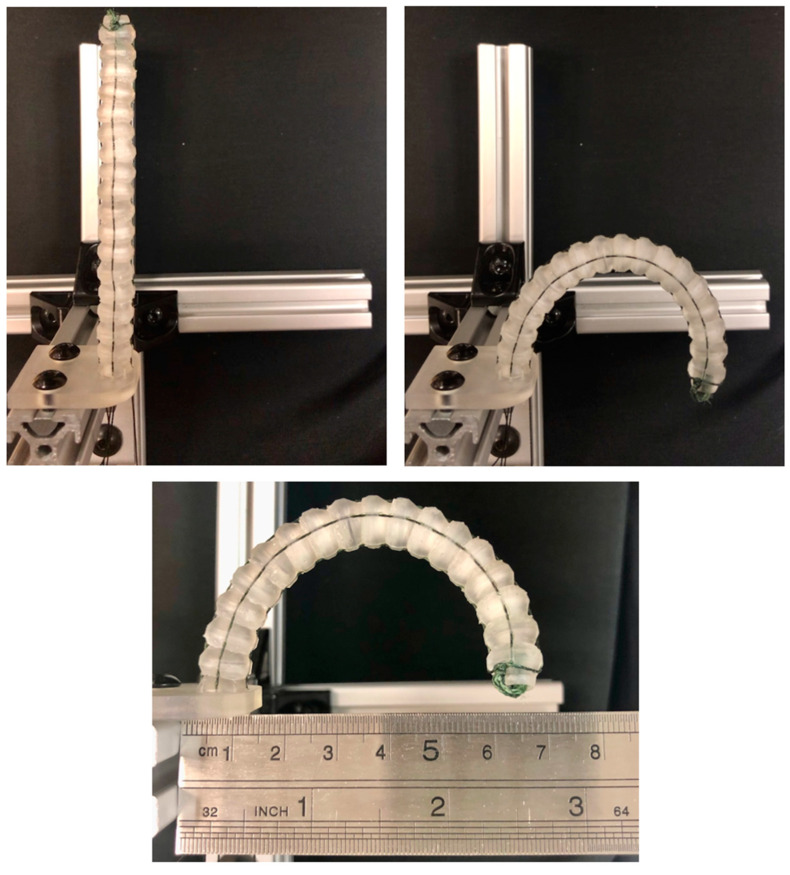
Photographs of experimental validations of Model 2 prototype: (**top left**) in undeflected state; (**top right**) in maximum deflection state; (**bottom**) with ruler showing a radius of curvature of approximately 40 mm.

**Table 1 bioengineering-11-00417-t001:** Design parameters: variable and fixed (designated by *).

Parameter	Description	Lower Bound	Upper Bound	Units
$R_{2}$	Spherical radius	3.00	6.00	mm
$H_{1}$	Flange height	2.00	12.00	mm
$H_{2}$	Flange chamfer height	1.00	11.00	mm
μ	Socket depth	1.00	6.00	mm
$D_{3}$	Central lumen diameter *	5.00	5.00	mm
$D_{5}$	Radial lumen diameter *	0.80	0.80	mm
c	Socket radial clearance *	0.03	0.03	mm
∆	Flange to spherical center dist.	0.01	3.00	mm
ψ	Flange chamfer angle	15.0	40.0	deg
λ	Central lumen chamfer angle	15.0	75.0	deg
$t_{i}$	Flange inner wall thickness *	0.10	0.10	mm
$t_{o}$	Flange outer wall thickness *	0.25	0.25	mm

**Table 2 bioengineering-11-00417-t002:** Optimal probe configurations for corresponding pulmonary nodule locations.

PN Number	Lobe	Port Number	$L_{p}$ (mm)	$L_{d}$ (mm)	$L_{tot}$ (mm)	Φ (deg)	ρ (mm)
1	RUL	1	163.21	99.48	262.69	64.86	41.98
2	RUL	2	45.04	120.59	165.63	37.49	57.10
3	RML	2	56.93	58.97	115.9	18.97	29.08
4	RML	1	128.62	17.93	146.55	98.07	5.88
5	RLL	4	67.75	29.47	97.22	46.25	13.55
6	RLL	4	38.09	30.76	68.85	53.72	13.72
7	LUL	5	37.24	120.03	157.27	27.97	58.23
8	LUL	5	109.07	144.15	253.22	69.97	59.05
9	LLL	6	51.17	25.05	76.19	38.78	11.81
10	LLL	7	39.44	13.18	52.62	86.78	4.79

**Table 3 bioengineering-11-00417-t003:** Pareto-optimal designs that resulted in minimal objective functions (highlighted in gray) for each trial.

Trial	Optimum	$R_{2}$	Δ	$H_{1}$	ψ	$H_{2}$	λ	μ	ρ	$I_{xx}$	$D_{1}$
1	Minimal ρ	5.14	1.28	6.18	54.07	2.29	20.12	4.75	19.94	39,321.51	13.20
1	Minimal $I_{xx}$	3.66	0.01	5.93	58.31	1.72	15.00	3.97	55.87	17,819.53	10.23
1	Minimal $D_{1}$	3.66	0.01	5.93	58.31	1.72	15.00	3.97	55.87	17,819.53	10.23
2	Minimal ρ	5.14	1.20	6.27	55.27	2.22	20.12	4.75	20.26	39,774.94	13.20
2	Minimal $I_{xx}$	3.66	0.01	5.93	58.31	1.72	15.02	3.96	55.82	17,822.57	10.23
2	Minimal $D_{1}$	3.66	0.01	5.93	58.31	1.72	15.00	3.97	55.87	17,819.53	10.23
3	Minimal ρ	4.89	1.15	5.18	41.11	2.49	18.08	4.64	18.83	33,678.06	13.00
3	Minimal $I_{xx}$	3.68	0.02	6.58	62.24	1.59	15.02	3.84	42.61	18,296.35	10.29
3	Minimal $D_{1}$	3.67	0.01	7.33	56.60	3.39	15.00	3.89	61.46	26,597.28	10.23

**Table 4 bioengineering-11-00417-t004:** MANOVA results for each design variable.

	Design Parameter						
Performance Index	$R_{2}$	Δ	$H_{1}$	ψ	$H_{2}$	λ	μ
Univariate *p*-value: ρ	<0.001	<0.001	<0.001	0.793	0.972	<0.001	<0.001
Univariate *p*-value: $I_{xx}$	<0.001	0.591	<0.001	0.000	<0.001	0.753	0.500
Univariate *p*-value: $D_{1}$	<0.001	<0.001	<0.001	<0.001	<0.001	0.587	0.025
Univariate ${\eta_{p}}^{2}$: ρ	0.023	0.128	0.171	0.000	0.000	0.445	0.100
Univariate ${\eta_{p}}^{2}$: $I_{xx}$	0.045	0.001	0.223	0.632	0.054	0.000	0.001
Univariate ${\eta_{p}}^{2}$: $D_{1}$	0.060	0.021	0.194	0.468	0.020	0.001	0.004

## Data Availability

All authors agree to share the research data presented in this paper.

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
