# Peer review of "Design Optimization and Tradeoff Analysis of an Actuated Continuum Probe for Pulmonary Nodule Localization and Resection"

_bioengineering, 2024, doi:10.3390/bioengineering11050417_

Round 1
Reviewer 1 Report
Comments and Suggestions for Authors
Gregory D. Buckner et al. reported an interesting work upon the design of a pulmonary surgery apparatus. The topic fell within the scope of Bioengineering, and could be considered for publication after a Minor Revision. Please refer to the following comments:
1. Some peripheral figures could be moved to Supplementary Files. There were too many figures in the main text.
2. Please demonstrate the data source of the human body configuration in Figure 2.2 ~ 2.5.
3. Please enlarge Figure 3.4 to make it clearer.
4. An individual Conclusion Section should be added.
Author Response
1. Some peripheral figures could be moved to Supplementary Files. There were too many figures in the main text.
The authors agree, and have moved Figures 3.6 (which consists of 21 subplots) and 3.7 (which consists of 7 subplots) to the Supplementary Materials section as Figures S1 and S2. The associated text now reads "To further investigate these relationships and the effects of other design variables, the variables of every design in the space are plotted against the three objective functions in Supplementary Materials Figure S1." These changes resulted in the re-numbering of subsequent figures.
2. Please demonstrate the data source of the human body configuration in Figure 2.2 ~ 2.5.
We have added a reference to the data source for Figures 2.2-2.5 in the caption of Figure 2.2 (reference [35]), in addition to the reference in the Methods and Materials section of the manuscript: "To establish kinematic and design requirements for the actuated probe, CT scan data of an adult human thoracic cavity [35] was used to create a point cloud model in MATLAB (Figure 2.2). This model was scaled based on published anthropometric measurements to provide the 40% deflation common in lung resections [36]. "
3. Please enlarge Figure 3.4 to make it clearer.
The authors agree that Figure 3.4 was too small to effectively convey the Pareto frontier information, and have made it larger and clearer in the revised manuscript.
4. An individual Conclusion Section should be added.
Bioengineering's "Instructions for Authors" notes that research manuscript sections should be "Introduction, Materials and Methods, Results, Discussion, Conclusions (optional)". This is the classic AIMRaD format typical of most medical journals. Bioengineering's Word Template further clarifies that "5. Conclusions: This section is not mandatory but can be added to the manuscript if the discussion is unusually long or complex."
The authors do not feel that our Discussion section is unusually long or complex, thus we request the omission of a Conclusion section. We would be happy to change the Discussion section title to either "Discussion/Conclusion" or simply "Conclusion" if the editor recommends such a change.
Reviewer 2 Report
Comments and Suggestions for Authors
The study presents a novel approach to enhancing the accuracy of margin delineation during wedge resection of pulmonary nodules through the development of a modified ultrasound imaging technique. The study is well written, with an elegant analysis of the proposed approach. However, I have a methodology concern:
- The study employs multiple objective genetic algorithms (MOGAs) and design of experiments (DOE) to explore the design space and quantify key dimensional relationships. However, it lacks a detailed discussion on the limitations and assumptions underlying these methodologies, raising questions about the robustness of the findings. I suggest to more detail the discussion section highlighting these possible study limitations.
Author Response
The study employs multiple objective genetic algorithms (MOGAs) and design of experiments (DOE) to explore the design space and quantify key dimensional relationships. However, it lacks a detailed discussion on the limitations and assumptions underlying these methodologies, raising questions about the robustness of the findings. I suggest to more detail the discussion section highlighting these possible study limitations.
The reviewer makes an excellent point and suggestion. We have revised text in the Discussion section to address these issues. Specifically, we have added the following:
"The computational tools used in this study, multiple objective genetic algorithms (MOGA) and design of experiments (DOE), have inherent assumptions and limitations that could affect the accuracy and robustness of the dimensional relationships. The inherent assumptions of continuous and convex objective spaces, the potential for convergence to local minima, and the unknown sensitivities of design outcomes to variations in design parameters could adversely impact the accuracy of results and conclusions presented here. This study was computationally limited by the maximum number of MOGA generations, population size, and number of trials conducted so in future work, varying crossover rates, mutation rates, population sizes and stopping criteria; all clearly affect the convergence of Pareto frontiers and complete exploration of the design space. Also, using improved methods for evaluating convergence of the multiple objective design optimization, such as hypervolume metrics, might be beneficial. A more sophisticated anatomical simulation could also be used to better quantify the probe actuation requirements by including additional constraints caused by the introduction of other laparoscopic instruments (staplers, cameras, etc.) into the surgical space. This simulation could also be used to observe if having multiple bending segments would be beneficial for probe navigation."